# Prospective surveillance study to detect antimalarial drug resistance, gene deletions of diagnostic relevance and genetic diversity of *Plasmodium falciparum* in Mozambique: protocol

Alfredo Mayor  ,[1,2,3,4] Clemente da Silva,[1] Eduard Rovira-Vallbona,[2] Arantxa Roca-Feltrer,[5] Craig Bonnington,[5] Alexandra Wharton-Smith,[5] Bryan Greenhouse,[6] Caitlin Bever,[7] Arlindo Chidimatembue,[1] Caterina Guinovart,[2] Joshua L Proctor,[7] Maria Rodrigues,[5] Neide Canana,[5] Paulo Arnaldo,[8] Simone Boene,[1] Pedro Aide,[1,8] Sonia Enosse,[5] Francisco Saute,[1] Baltazar Candrinho[9]

**Correspondence to**
Alfredo Mayor;
alfredo.mayor@isglobal.org

## ABSTRACT

**Introduction** Genomic data constitute a valuable adjunct to routine surveillance that can guide programmatic decisions to reduce the burden of infectious diseases. However, genomic capacities remain low in Africa. This study aims to operationalise a functional malaria molecular surveillance system in Mozambique for guiding malaria control and elimination.

**Methods and analyses** This prospective surveillance study seeks to generate *Plasmodium falciparum* genetic data to (1) monitor molecular markers of drug resistance and deletions in rapid diagnostic test targets; (2) characterise transmission sources in low transmission settings and (3) quantify transmission levels and the effectiveness of antimalarial interventions. The study will take place across 19 districts in nine provinces (Maputo city, Maputo, Gaza, Inhambane, Niassa, Manica, Nampula, Zambézia and Sofala) which span a range of transmission strata, geographies and malaria intervention types. Dried blood spot samples and rapid diagnostic tests will be collected across the study districts in 2022 and 2023 through a combination of dense (all malaria clinical cases) and targeted (a selection of malaria clinical cases) sampling. Pregnant women attending their first antenatal care visit will also be included to assess their value for molecular surveillance. We will use a multiplex amplicon-based next-generation sequencing approach targeting informative single nucleotide polymorphisms, gene deletions and microhaplotypes. Genetic data will be incorporated into epidemiological and transmission models to identify the most informative relationship between genetic features, sources of malaria transmission and programmatic effectiveness of new malaria interventions. Strategic genomic information will be ultimately integrated into the national malaria information and surveillance system to improve the use of the genetic information for programmatic decision-making.

**Ethics and dissemination** The protocol was reviewed and approved by the institutional (CISM) and national

## STRENGTHS AND LIMITATIONS OF THIS STUDY

⇒ Next-generation sequencing will be performed in country through the establishment of technical and computational infrastructure as well as analytical tools.

⇒ The project builds from recent elimination experiences in southern Mozambique and uses a biorepository of already collected *Plasmodium falciparum* samples to select multiallelic short-range haplotypes (microhaplotypes) that increase the power of biallelic loci for phase inference in polygenomic infections.

⇒ A joint epidemiological-genetic analysis will enable better predictions of the operational efficacy of new interventions.

⇒ We will assess the value of a new surveillance system at antenatal visits to improve the programmatic performance of malaria control and elimination activities.

⇒ More evidence on the association between malaria transmission intensity and genetic data is required for the use of malaria molecular surveillance data to assess the effectiveness of malaria interventions.

ethics committees of Mozambique (Comité Nacional de Bioética para Saúde) and Spain (Hospital Clinic of Barcelona). Project results will be presented to all stakeholders and published in open-access journals.

**Trial registration number** NCT05306067.

## INTRODUCTION

Pathogen genomics has the potential to transform the surveillance, prevention and control landscape of infectious diseases. The rapid innovation in sequencing technologies has led to the development of robust

next-generation sequencing equipment with the ability for high pathogen resolution at increasingly affordable prices. This development has subsequently facilitated the incorporation of pathogen genomics in disease surveillance systems in high-income countries, allowing for targeted and effective control of disease threats through the timely and in-depth pathogen characterisation.[1] Genomics-based surveillance is therefore becoming an integral strategy towards control and elimination of diseases such as COVID-19, tuberculosis, malaria, HIV and food-borne pathogens, among others.[2]

The strategic use of genetic variation in *Plasmodium falciparum* can boost the capacity of malaria control and elimination programmes to deploy the most efficient interventions.[3] Molecular tools and use cases for decision making are currently being considered by the WHO which, through a technical consultation on the role of parasite and anopheline genetics in malaria surveillance,[4] identified different levels of action based on evidences available. Genetic data can flag the emergence of mutations conferring resistance to antimalarials (ie, artemisinins)[5] or deletions that affect rapid diagnostic test (RDT) sensitivity (ie, *P. falciparum histidine-rich protein 2* [*pfhrp2*]).[6–8] Genomic scans for selection[9] can identify other parasite adaptations mediated by single nucleotide polymorphisms (SNPs) and structural variations (gene copy number)[10] that may require a programmatic response. Parasite-relatedness metrics such as identity by descent (IBD)[11] can be used to characterise the key drives of ongoing transmission to identify foci[12 13] and to discriminate between indigenous and imported cases in areas approaching elimination.[14–16] Bottlenecks in parasite population driven by control and elimination efforts have been shown to reduce *P. falciparum* genetic diversity and increase similarity due to inbreeding and recent common ancestry.[17] These evidences provide the basis for modelling efforts to recapitulate features of malaria transmission from genetic data and inform about the effectiveness of antimalarial interventions.[18–23] However, further evidence is needed to demonstrate the feasibility and appropriateness of using genetic data as a proxy for transmission intensity and define the conditions under which that feasibility applies. Moreover, standardised approaches for detecting resistance through molecular markers are lacking, and variation in sample type, collection, storage, DNA extraction, marker detection and analysis of results can undermine the comparability of findings, as well as the sensitivity and specificity of methods used. Adequate genotyping methods, sampling frameworks, analytical pipelines and demonstration studies are still required across a range of malaria intensities, programmatic environments and use scenarios.

Strategic *P. falciparum* genetic information can be integrated into innovative cost-efficient surveillance approaches, such as those targeting pregnant women attending antenatal care (ANC) clinics.[24] Women at ANC are a generally healthy, easy-access population, contributing valuable data for infectious disease surveillance (ie, HIV[25] and syphilis[26]) and wider health metrics at the community level, including a proxy of the malaria burden in the community.[27–32] Moreover, ANC-level malaria surveillance can provide a routine measure of the malaria burden in pregnancy, which countries lack, while potentially improving pregnancy outcomes by treating infections at first trimester. Women attending ANC also provide an attractive sampling population for measures of exposure to malaria beyond simply presence or absence of parasite infection. In particular, in addition to measuring complexity of infection or parasite flow-rates between populations, molecular analysis of *P. falciparum* samples collected from pregnant women may provide a means for the identification of adaptations developed by the parasite to control strategies, such as antimalarial resistance and deletions of antigens targeted by RDT that can compromise diagnosis, treatment and prevention.

Despite the potential benefits and the greater need to control the high burden of infectious diseases, genomic surveillance capacity remains low for many public health programmes in Africa.[2] In order to reduce inequities in the access to sequencing technologies, this project aims to promote capacities in Mozambique for operationalising a functional malaria molecular surveillance (MMS) system for decision-making.[4] Mozambique is among the 10 countries with the highest burden of malaria worldwide, with an estimated 10.8 million cases in 2020.[33] However, malaria transmission is very heterogeneous in the country, with a high burden in the north and very low transmission in the south. Therefore, the project aims to address National Malaria Control Programme (NMCP) programmatic needs for elimination initiatives in southern Mozambique and burden reduction in the north (figure 1).

## METHODS AND ANALYSIS
### Study design
This is a prospective genomic surveillance study of *P. falciparum* samples to be collected between 2022 and 2023 from a variety of transmission intensities and geographies in Mozambique to inform three use cases: appropriate malaria diagnostics and treatment; characterising transmission sources in low transmission settings and identifying intervention mixes with optimal effectiveness to reduce burden in moderate-to-high transmission areas. To achieve this, three different sampling approaches will be performed. First, all malaria cases will be sampled throughout the year in two low transmission districts of southern Mozambique currently targeted by reactive malaria surveillance activities (*dense sampling*). Second, a targeted approach will aim to collect a predefined number of samples at selected health facilities in the country. In low transmission settings, sampling will be conducted throughout the year, while two surveys will be conducted in medium-to-high transmission settings: one during the rainy and a second one during the dry season (which extend from November to April and May to October,

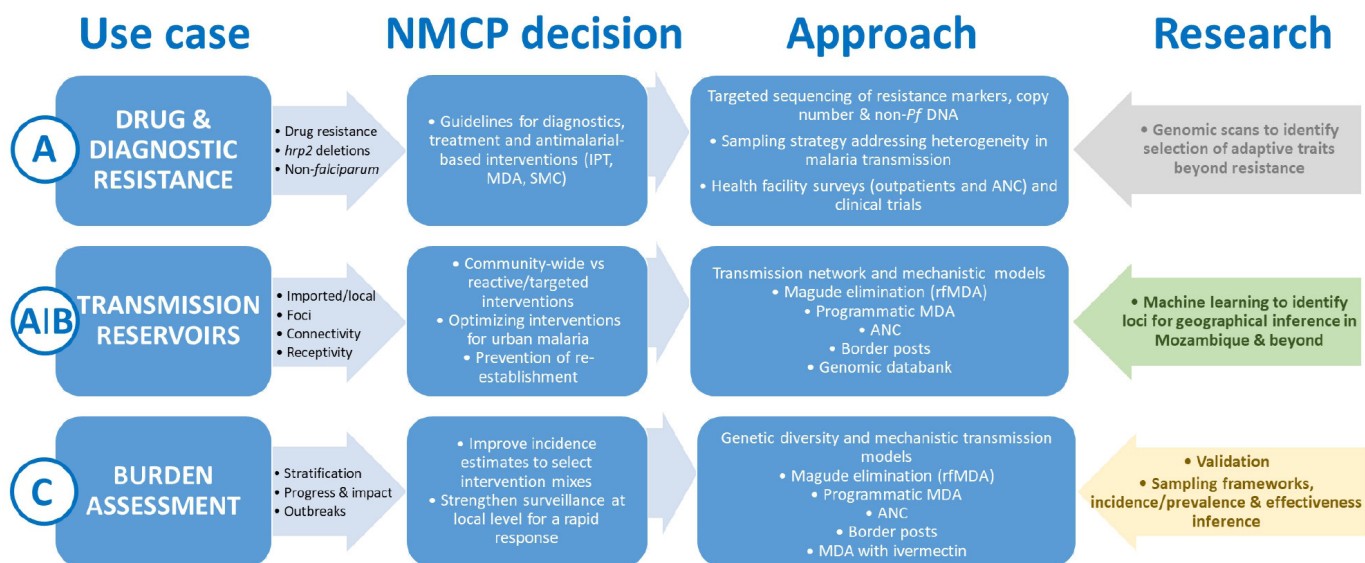

**Figure 1** Malaria genomic use cases and National Malaria Control Programme (NMCP) decisions. The letter on the left (A–D) expresses the level of action described in the WHO Technical consultation on the role of parasite and anopheline genetics in malaria surveillance. (A) Immediate action; (B) medium-term action; (C) long-term action. Arrows in colour at the right express the research required for action in the medium-term and long-term (grey, not essential for action; green, immediate evidence; yellow, medium-term evidence). ANC, antenatal care clinics; IPT, intermittent preventive treatment; MDA, mass drug administration; rfMDA, reactive focal MDA; SMC, seasonal malaria chemoprevention.

respectively). During the high transmission (rainy) season, an LDH-based RDT will be added to the standard routine HRP2-based diagnostics to identify potential false negative results due to *pfhrp2/3* deletions among clinical cases.[34] And third, *ANC sampling* of pregnant women at first attendance will be conducted throughout the year at selected health facilities in the country. The overarching sampling strategy for the study will however remain flexible and iterative, informed by sample analysis as the study progresses, and in view of future sampling and research activities being conducted by the Ministry of Health, National Institute of Health and other stakeholders in Mozambique, to avoid sampling overlap and ensure a diversity of sampled locations.

The project will also leverage from clinical trials and surveillance activities being conducted in Mozambique between 2021 and 2024, namely the Malaria Indicator Survey (2022–2023) in southern Mozambique; the therapeutic efficacy survey (2022) in sentinel sites in the country (Montepuez in Cabo Delgado, Moatize in Tete, Dondo in Sofala, Mopeia in Zambézia and Massinga in Inhambane);[35 36] reactive surveillance activities in Magude and Matutuine (Maputo Province; 2022-2023); a phase III cluster-randomised, open-label, clinical trial in 2022 to study the safety and efficacy of ivermectin mass drug administration to reduce malaria transmission in Mopeia District (Zambézia Province); a large-scale implementation development project aiming at maximising the delivery and uptake of perennial malaria chemoprevention (formerly intermittent preventive treatment in infancy) in Massinga District (Inhambane Province; 2022–2024); a hybrid effectiveness-implementation study to evaluate the feasibility and effectiveness of seasonal malaria chemoprevention with sulfadoxine–pyrimethamine (SP) and amodiaquine in Nampula Province (2022) and a programmatic delivery of a population-based mass drug administration with dihydroartemisinin-piperaquine in Manjacaze district (Gaza Province; 2022–2023).

## Study settings and participants

Nine provinces were identified through consultation with the NMCP for inclusion in the study: Maputo City, Maputo, Gaza, Inhambane, Niassa, Manica, Nampula, Zamb'ézia and Sofala. Selection of study sites will be stratified by transmission intensity into two major strata: (a) low transmission (Maputo city and Maputo Province, where individual case notification is being implemented to reach interruption of transmission) and (b) medium-to-high transmission areas (Gaza, Inhambane, Niassa, Manica, Nampula, Zamb'ézia and Sofala provinces, targeted by burden-reducing strategies). Overall, a total of 19 districts will be included, which provide a diverse range of epidemiological settings (see table 1 and figure 2).

Dense sampling will be conducted in the low transmission districts of Magude and Matutuine (Maputo Province), where all the individuals of any age (>6 months old) with clinical symptoms of malaria (defined as axillary temperature ≥37.5°C or history of fever in the preceding 24 hours) and a parasitologically confirmed malaria diagnosis via RDT or microscopy (table 2) will be invited to donate their RDT for molecular analysis (dense sampling).

Targeted sampling will be conducted at selected health facilities in the low transmission districts of Boane and Manhiça (Maputo Province ), and KaMavota, KaMaxaqueni and Nhamankulu Districts (Maputo City), where

**Table 1** Study provinces and districts targeted in the protocol

| Transmission | Region | Province | District | Dense | HFS | ANC | Other sources |
|---|---|---|---|---|---|---|---|
| | | | | | Sampling | | |
| | | | | | Targeted | | |
| Low | South | Maputo City | Kamavota, KaMaxaqueni and Nlhamankulu | | X* | | |
| | | Maputo Province | Boane and Manhiça | | X* | | |
| | | | Magude | X* | | X† | React |
| | | | Matutuine | X* | | | React |
| Medium-to-high | | Gaza | Manjacaze | | X‡ | X† | MDA-DP |
| | | Inhambane | Maxixe | | X‡ | X† | |
| | | | Massinga | | X‡ | X† | PMC and TES |
| | Central | Manica | Guro and Gondala | | X‡ | X† | |
| | | Sofala | Chemba | | X‡ | X† | |
| | | | Dondo | | | | TES |
| | | Tete | Moatize | | | | TES |
| | North | Niassa | Cuamba | | X‡ | X† | |
| | | Nampula | Mecuburi, Malema, Lalaua and Muecate | | X‡ | X† | SMC |
| | | Zambézia | Mopeia | | X‡ | X† | MDA-IVM and TES |
| | | Cabo Delgado | Montepuez | | | | TES |

*Year round, all ages.
†Year round, first ANC visit.
‡Rainy and dry season; 2–10 years of age.
ANC, antental care clinics; HFS, health facility survey; MDA-DP, mass drug administration with dihydroartemisinin-piperaquine; MDA-IVM, mass drug administration with ivermectin; PMC, Perennial malaria chemoprevention; React, reactive surveillance; SMC, seasonal malaria chemoprevention; TES, therapeutic efficacy study.

a drop of blood will be collected onto filter paper from consenting individuals of any age (>6 months old) with confirmed clinical malaria. In medium-to-high transmission areas, targeted sampling will focus on children aged 2–10 years of age attending selected health facilities with clinical symptoms of malaria and a parasitologically confirmed malaria diagnosis via RDT (table 2). Ten health facilities will be targeted in each district.

Pregnant women attending their first antenatal care visit (any trimester) will be invited to participate both in low (Maputo Province) and high transmission provinces (Inhambane, Gaza, Nampula, Niassa, Manica, Sofala and Zambézia; table 1), irrespectively of malaria clinical symptoms.

### Enrolment of participants

Dense sampling in Magude and Matutuine districts will be coordinated with district malaria focal points, community health workers (CHW), malaria volunteers (who provide a link between the CHW and the health facility, and assist the CHW in the follow-up of cases and administration of medication) and health facilities. All *P. falciparum* positive RDTs (i.e., SD Bioline Malaria Ag Pf, 05FK50, Abbott) will be stored for molecular analysis.

RDTs of *P. falciparum*-confirmed household contacts will also be collected to estimate the rate of within-household transmission. Targeted sampling through health facility-based surveys (HFS) in low and medium-to-high transmission settings will be carried out by one team comprised of one maternal and child health nurse, a laboratory technician or a medical technician. The number of people to be screened in each health facility and the duration of recruitment to achieve the sample size will be dependent on the RDT-positivity rate among people meeting the eligibility criteria. A second test including a non-HRP2 line (StandardQ Malaria Pf/Pan Ag Test, SD Biosensor) will be carried out in HFS during the rainy season and discrepant results suggestive of *pfhrp2/3* deletions will be recorded and further analysed to confirm the deletion. Nurses at the ANC clinics will be in charge of the recruitment of pregnant women at their first visit. Pregnant women will be tested for malaria using a routine RDT and the result will be recorded in a standard questionnaire, together with routine ANC tests. Each enrolled individual will be assigned with a unique identification (UID) number and a barcode.

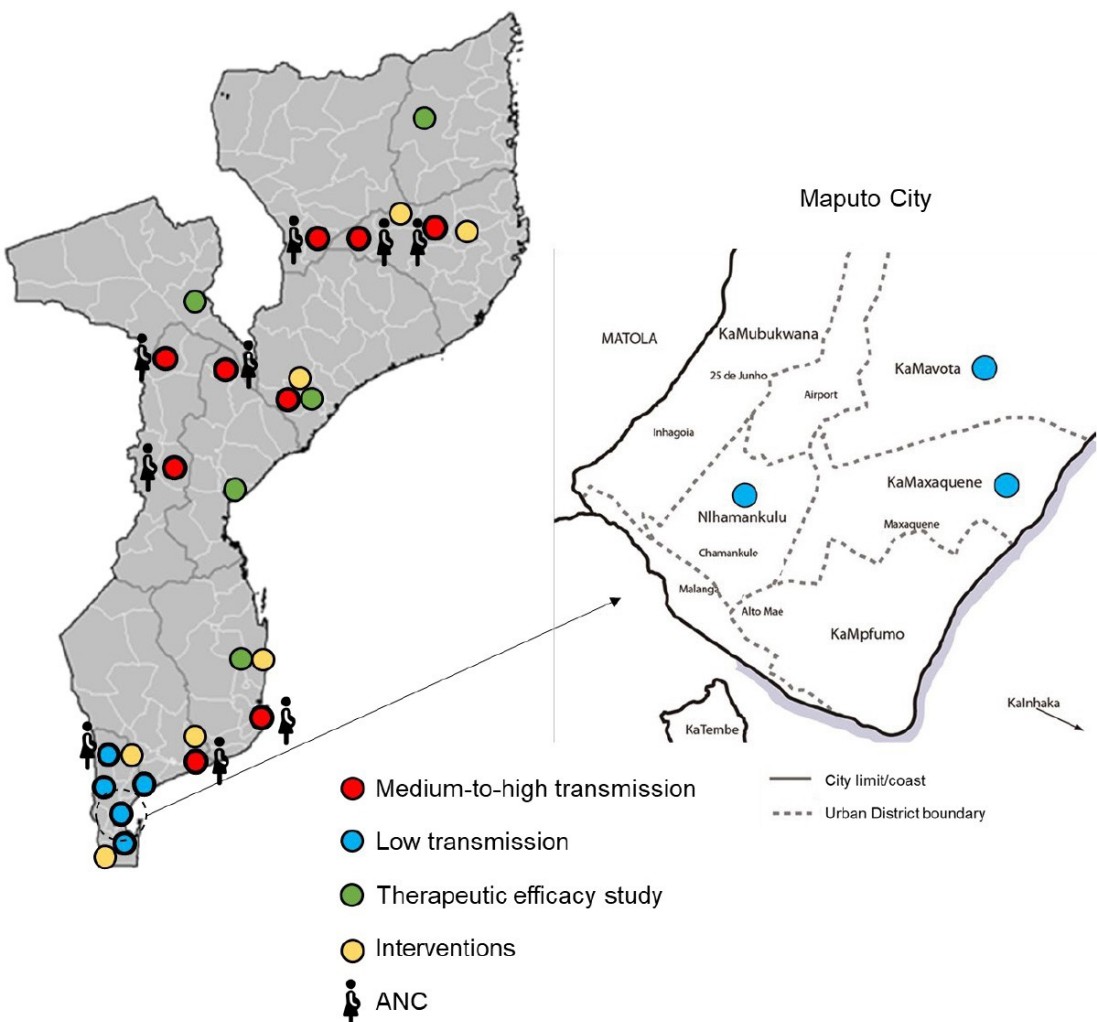

**Figure 2** Low and medium-to-high transmission study districts targeted in the protocol. ANC, antental care clinics.

## Data and sample collection

Field workers and nurses will be trained to ask for informed consent (online supplemental appendix 1–4), perform a simple questionnaire (online supplemental appendix 5–8) and collect biological samples for molecular analysis. The survey questionnaire will be administered to all study participants or children's parents/guardians meeting the inclusion criteria and will include inclusion criteria check, characteristics of the participant and malaria-related information. For pregnant participants, data will be collected on parity and gestational age at first ANC visit, as well as information related with malaria and use of preventive measures. A telephone contact number will be collected from pregnant women in low transmission settings in order to locate their residence for spatial analysis. In areas targeted by reactive surveillance activities (Magude and Matutuine in Maputo Province), travels during the previous 30 days to the case notification will be registered, including destinations and dates. A Site Coordinator will be responsible for supervising the work of field workers, nurses and the data entry clerk, and for reviewing and comparing questionnaires and samples for correct matching, completeness and accuracy.

Nurses will be trained to collect blood by finger pricking (online supplemental table 1) following standard (online supplemental appendix 9) and COVID-19 safety procedures (online supplemental appendix 10). For each participant, either the *P. falciparum*-positive RDT used for routine malaria diagnosis (dense sampling) or four blood spots onto two filter papers (Whatman Grade CF 12; targeted sampling) will be collected. Specimens will be labelled anonymously (patient UID, study health facility and date), dried for 24 hours and kept in individual plastic bags with desiccants at 4°C. Every 2–6 weeks, the completed questionnaires, informed consents and samples will be sent to the data entry clerk at CISM through a local transportation agency. Informed consents will be received by study investigators. A data manager will be responsible for the receipt of the informed consents and double data entry at CISM, and a laboratory technician will be responsible for receiving the samples and store them at −20°C until analysis. Part of the dried blood spot will be stored in RNA-preserving solution. All samples will be kept in the CISM laboratory for a period of approximately 15 years. For quality control purposes, up to 5% of the samples will be analysed at UCSF (San

**Table 2**  Study eligibility criteria

| Inclusion criteria | Exclusion criteria |
|---|---|
| **Low transmission Provinces** | |
| ▶ > 6 months | ▶ Any symptoms of severe malaria |
| ▶ Fever (axillary temperature ≥37.5°C) or history of fever in the preceding 24 hours | ▶ Negative parasitological test for malaria via RDT or microscopy (except any women at their first ANC visit, who will be recruited before testing for malaria with an RDT) |
| ▶ Positive parasitological test for malaria diagnosis via RDT or microscopy | ▶ Unwilling to provide informed, written consent |
| ▶ Household contact of someone with fever/history of fever and *Plasmodium falciparum* positive RDT | ▶ History of antimalarial treatment in the last 14 days |
| OR | |
| ▶ Pregnant women attending first antenatal care visit in Magude district | |
| AND | |
| ▶ Informed, written consent to participate from participant and/or guardian | |
| **High transmission Provinces** | |
| ▶ Children aged 2–10 years of age | ▶ Any symptoms of severe malaria |
| ▶ Fever (axillary temperature ≥37.5°C) or history of fever in the preceding 24 hours | |
| ▶ Positive parasitological test for malaria diagnosis via RDT* or microscopy | ▶ Negative parasitological test for malaria via RDT or microscopy (except any women at their first ANC visit, who will be recruited before testing for malaria with an RDT) |
| OR | |
| ▶ Pregnant women attending first antenatal care visit | ▶ Unwilling to provide informed, written consent |
| AND | ▶ History of antimalarial treatment in the last 14 days |
| ▶ Informed, written consent to participate from participant and/or guardian | |

*A second RDT (HRP2-pLDH) will be provided in these locations to support detection of *P. falciparum hrp2* deletions.
ANC, antenatal care; RDT, rapid diagnostic test.

Francisco, USA) and/or ISGlobal (Barcelona, Spain). In order to identify errors in data or sample collections and take necessary corrective actions, a standardised checklist (online supplemental appendix 11) will be filled in by the monitoring officer during biweekly monitoring visits.

## Molecular analyses

Informative SNPs (including—but not restricted to—markers of resistance to artemisinin [*pfkelch13*],[37] SP [*pfdhfr, pfdhps*],[38] or chloroquine (*pfcrt*)[39]), microhaplotypes[40] and *pfhrp2* and *pfhrp3* regions[6–8] will be targeted using multiplexed primers on flanking sequences, with a range of amplicon size of ~225–275 bp (covered by a paired end read). Targeted amplicons obtained by PCR on genomic DNA using Illumina-specific adaptors and sample-specific barcode will be pooled to create a single product library, which will be sequenced (paired-end 150 bp) on a Miseq Illumina sequencer in the country or higher performing equipment when available. Amplicon representation and SNP and haplotype calling will be assessed in demultiplexed and trimmed sequencing reads after filtering sequencing errors. The designed panel will

be validated using mixtures of *P. falciparum* lines to determine precision and repeatability. Genotyping methods, including number of SNPs and microhaplotypes to be characterised, distribution across the parasite's chromosomes, the proportion of putatively neutral versus non-neutral polymorphisms, pooling strategy and criteria for validating sequencing data (ie, minimum sequencing depth and maximum error rate) will be developed as part of this project. Samples will also be used for other molecular analysis of programmatic interest, such as the detection of *Plasmodium* species, parasite antigens, serological markers of parasite exposure (antibodies) and parasite RNA-based markers (ie, gametocytes). A quality control programme based on the sequencing of an artificially created set of samples (ie, mixtures of known laboratory controls at specific proportions and densities) will be processed at predefined times to guarantee the quality of the processes during the life of the project.

## Data management

Data will be collected using paper (targeted sampling) and password-protected electronic devices (dense sampling).

  Mayor A, *et al. BMJ Open* 2022;**12**:e063456. doi:10.1136/bmjopen-2022-063456

Data collected using paper will be double entered into the study database using RedCap.[41] Automatic quality checks will be performed to ensure data completeness. Confidentiality and security will be ensured through automatic encryption of sensitive data, storage in password-protected computers and locked locations, and data sharing using password-protected, encrypted files. Prior to analysis, data will be deidentified with the exception of geolocation codes, which are necessary for specific analyses. The study will also use data available from the NMCP, including intervention coverage, historical prevalence surveys, travel history or other mobility assessments and entomological data. Sequences generated through the analysis of samples will be integrated into a curated catalogue of genomic data together with relevant anonymised clinical and epidemiological information and will be made publicly available in public repositories such as the European Nucleotide Archive (ENA) and MalariaGen Resource Centre. In order to facilitate data accessibility and use, and to obtain a meaningful integration with other sources of surveillance data, genetic information will be incorporated into the DHIS2-based Integrated malaria information storage system (iMISS), which is currently being rolled out in Mozambique.[42]

## Study outcomes and sample size calculations

The primary endpoints are as follows: (a) prevalence of molecular markers of diagnostic and antimalarial resistance by period, study area and population (use case 1); (b) genetic-relatedness indicators between pairs of samples and populations by period, study area and population (use case 2) and (c) genetic diversity indicators by period, study area and population (use case 3). Sample size per sampling domain (Province) has been estimated considering antimalarial and diagnostic resistance as a primary use case, considering the negligible carriage of molecular markers of artemisinin resistance[5] and *pfhrp2/3* deletions[6] in Mozambique, and setting 5% as the warning threshold.[43] Assuming a 10% of loss of samples or uninterpretable analysis, a sample size of up to 500 per sampling domain would be adequate to: (a) estimate a proportion of 0.05 (markers of drug resistance or *pfhrp2* deletion) with 0.026 absolute precision and 95% confidence and (b) achieve a power of 80% for detecting an increase of genetic marker (resistance or deletion) from 0 ‰ to 5% at a two-sided p-value of 0.01. A flexible and adaptive sampling scheme will be followed, where (a) estimates generated during the first half of the project will inform subsequent sampling schemes and (b) not all the samples collected will be analysed (some of them will be stored as reference materials, for confirmation of findings or future studies on *Plasmodium* biology). The number of pregnant women to be recruited in order to reach the sample number will depend on the parasite rates in the study areas; assuming an overall RDT positivity rate of 25%, we expect we will be needing to recruit a total of 2000 pregnant women per site to get 500 *P. falciparum*

positive samples, although numbers may differ between sites.

## Analysis plan

Demographic and clinical charateristics of study participants will be described using summary statistics. A user-friendly and locally executable bioinformatic pipeline will be developed for analysis of *P. falciparum* targeted sequencing data. Highly informative SNPs and microhaplotypes showing geographic structuring will be selected using a supervised machine learning approach trained by genomes from known geographic origin in Mozambique. Population-level genetic diversity will be quantified using expected heterozygosity (He), number of alleles per locus, allele frequency, complexity of infection (COI)[23] as well as other genetic metrics. Deletions and copy number variations will be assessed based on sequencing coverage ratios.[10 44] Methods to be used for population genetic analysis (including the genetic connectivity among infections, use of all vs only neutral SNPs, treatment of multiple-clone infections and integration of genetic data with travel history data) will be developed during the project. We will use regression models adjusted by potential confounders (demographic and clinical factors, among others) to compare genetic metrics between seasons, before and after the antimalarial interventions, between pregnant women and community sampling populations and across different intensities of malaria transmission. Finally, we will integrate genomic surveillance data into epidemiological and transmission network models. For the first one, we will leverage two recent models developed at the Institute of Disease Modelling[45] (a malaria genetic model calibrated to a longitudinal genetic study in Senegal[18] and a disease transmission model calibrated with the Magude data) to build an end-to-end malaria transmission and genetics model for Mozambique (figure 3). The transmission network model will include data for densely sampled in low transmission areas on individual and community-level case classification (imported, local and introduced), the extent and duration of sustained local transmission and how these change over space and time. Summary indicators will be visualised in graphical and tabular forms in the iMISS through genetic dashboards. We will establish risk profile algorithms and interpretation components that are capable of generating outputs on (a) country-wide antimalarial resistance profiles (rolling-basis); (b) in very low transmission areas (eg, Magude district), genetic connectivity and case classification (together with travel history and other parameters obtained from case-based notification tools) and (c) high burden to high impact specific analyses (ie, stratification and trend investigation for exploring the potential impact of intervention mixes implemented).

## ETHICS AND DISSEMINATION

The protocol was reviewed and approved by the institutional (CISM) and national ethics committees of

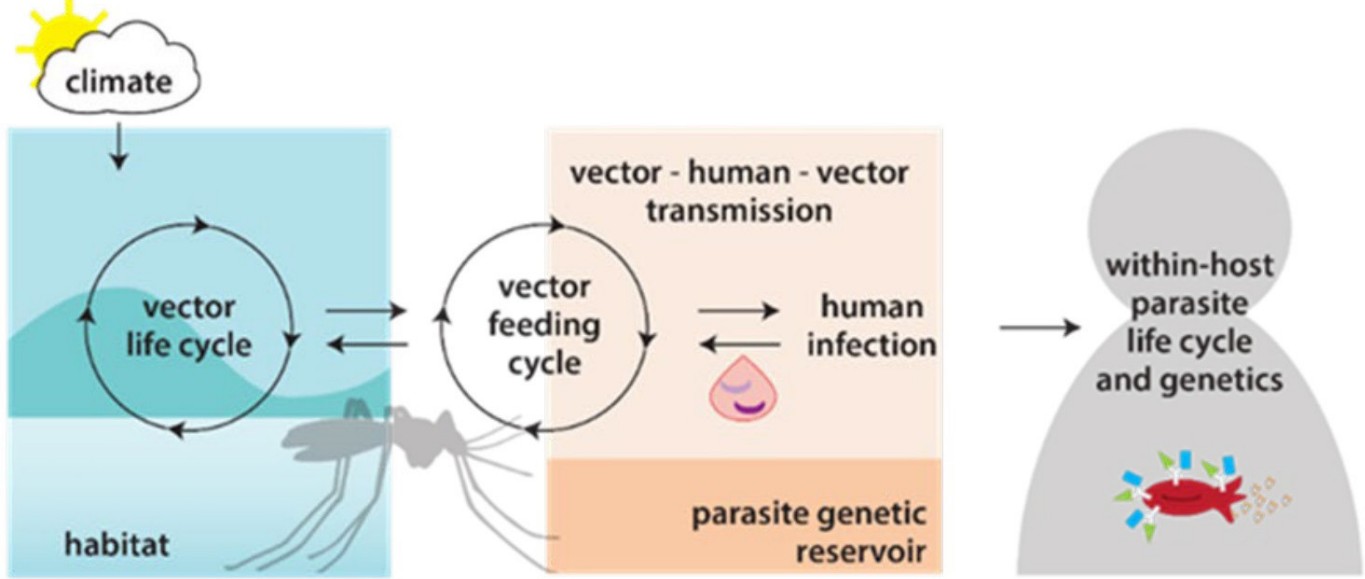

**Figure 3** Modelling approaches for malaria genomics. Overview of the components of a joint malaria epidemiology-genetic model, that builds on the capabilities of two models previously developed at the Institute of Disease Modelling (a malaria genetic model calibrated to a longitudinal genetic study in Senegal and a disease transmission model calibrated with the Magude data).

Mozambique and the Hospital Clinic of Barcelona. Written informed consent will be sought from all study participants before blood sample collection is conducted (online supplemental appendix 1). Two copies will be signed, one will be kept by participant and the other by the investigators in a locked space. The information sheet and consent form will also include text explaining informed consent for future use of biological specimens to conduct additional analyses of the *Plasmodium* parasite. In case of minors (less than 18 years of age), consent will be sought from parents, relatives or guardians. Informed consents will specify that the data will be made public. First-line treatment for malaria will be provided to the enrolled participants in line with national treatment guidelines. Considerations related to preventing the risk of SARS-COV-2 transmission are detailed in online supplemental appendix 10. There will not be any economic incentive to participate in the study. Transference of data and materials out of Mozambique will be done only when appropriate data and material transfer agreements are signed between participating institutions (online supplemental table 2).

### Patient and public involvement
Patients and the public were not involved in the development of this protocol.

### DISCUSSION
There is a growing acceptance that genomics can play a critical role in policy and programmatic decisions. With the aim of demonstrating the programmatic application and feasibility of malaria genomic surveillance in Mozambique, we will generate parasite genomic data across varying transmission scenarios for supporting strategic decision-making. First, MMS data will inform drug and diagnostic choices through the monitoring of molecular markers of antimalarial and diagnostic resistance. The emergence of *pfhrp2/3* deletions,[6–8] resistance to artemisinin[37] and partner drugs, as well as the resistance to SP used for chemoprevention,[38 46 47] threatens the global effort to reduce the burden of malaria.[33] The WHO recommends that countries with reports of *pfhrp2/3* deletions, and neighbouring countries, should conduct representative baseline surveys among suspected malaria cases. If the prevalence of molecular markers of antimarial resistance or deletions causing false negative RDT results reaches the threshold of >5%, then there is need to consider alternative antimalarials and RDTs. Second, the project will help to target the reservoirs sustaining transmission by quantifying parasite importation, identifying sources and characterising local transmission in near-elimination settings.[48 49] Genomic surveillance and phylogenetic analyses have enabled the near real-time estimation of transmission chains of non-sexually recombining, rapidly evolving pathogens such as Ebola,[50] influenza[51] and COVID-19.[52] However, molecular and analytic advancements are still required to characterise transmission patterns of pathogens such as *P. falciparum* with a sexually recombining stage.[49] Third, the project will assess the value of *P. falciparum* genetic diversity measures to supplement traditional surveillance for improving stratification, monitoring and impact evaluations in different epidemiological contexts, especially where surveillance data are sparse. This use case still requires development of analytical and interpretative to infer malaria burden[18 20 53–58] and effectiveness of interventions,[18–23 53 59–61] as well as

validation of sampling frameworks.[4] Finally, the project will test if parasite populations within pregnant women are representative of the general population and expand the usefulness of this approach to inform genomic surveillance indicators.

The project will use state-of-the-art sequencing and modelling approaches. Current *P. falciparum* genetic markers based on biallelic SNPs have limited support for polyclonal samples, which are frequent across all transmission intensities, and have limited resolution to calculate genetic relatedness between parasites, to estimate allele frequencies,[23 62] or to distinguish geographic origin.[21 23 63] Multiallelic short-range haplotypes (microhaplotypes) covered by a single read from high-throughput DNA sequencers allow an accurate statistical inference of phase and have the potential to derive more precise information than biallelic loci,[64–66] particularly in polyclonal infections, to tailor the genomic tool to specific transmission and geographic settings. In addition to being useful for identification and lineage/family relationships, microhaplotypes can provide information on biogeographic ancestry and can be useful for strain detection and deconvolution.[64–67] Methods such as IBD[11 68 69] that can exploit the signal left by recombination on these microhaplotypes may have the power to detect geographic differentiation at small spatial scales relevant for malaria control programmes. Machine learning approaches[70] will be used for the selection of key SNPs and microhaplotypes that allow accurate inference of malaria transmission and geographical origin. Finally, models that integrate genomic and epidemiological data will be developed to assess the programmatic effectiveness of new malaria interventions and characterise sources of malaria transmission (imported vs local).[45]

This project, guided by programmatic priorities and based on collaborative efforts, aims to boost the use of the genetic data for decision-making. To successfully achieve this, the project is grounded on three main principles: (a) strengthen sequencing capacities to implement a robust MMS system; (b) strong partnership and coordination to make MMS data sharing common practice for malaria control and elimination and (c) effective operationalisation of MMS implementation activities. Technical capacities will be built by establishing at CISM a sequencing platform and ancillary equipment for library preparation and quality control. Computational infrastructure and analytical tools will also be developed by establishing a user-friendly automated platform to analyse genomic data with simplified interpretation into actionable information. Training activities will target molecular biologists for wet laboratory analysis, a bioinformatician and molecular epidemiologists for data analysis and interpretation and a field epidemiologist for interpretation of the generated data, and public health specialists for adoption of the findings into policy. Genetic data-to-action culture and engagement of NMCP on genetic analysis will be promoted by integrating genetic aspects in the NMCP activities (ie, data review meetings) as well as in training and annual meetings, by integrating genetic information with other surveillance data onto the iMISS, and by documenting all the processes, successes and failures to inform future molecular activities. The project will pursue the use of MMS data as an adjunct to traditional surveillance information for elimination initiatives in southern Mozambique and burden reduction in the north through the engagement with regional malaria elimination initiatives (eg, E8 and MOSASWA, a trilateral initiative to eliminate malaria from Mozambique, South Africa and Eswatini[71 72]) and linking decision-making with the 'high burden to high impact' initiative under the guidance of WHO.

We expect that the genomic intelligence developed through this project will complement current and new surveillance systems to drive decision-making for the control and eventual elimination of malaria in Mozambique and other malaria endemic countries. However, further steps are required beyond this 3-year project. Enabling policies and regulatory mechanisms for sample storage and sharing,[73] adequate procurement of materials and infrastructure, as well as local expertise for equipment installation and maintenance, need to be developed for an effective integration of genomic surveillance into public health. Countries, with appropriate support from mainstream funding bodies, should also develop sustainability plans as part of national disease control programmes, emergency responses and other surveillance programmes (ie, antimicrobial resistance) to ensure resources for genomic surveillance. Finally, regular assessments of the efficiency and effectiveness of incorporating genomic data in routine public health surveillance systems will be crucial to stimulate the use of genetic data for policy making.

**Author affiliations**
[1]Centro de Investigação em Saúde de Manhiça, Manhiça, Maputo, Mozambique
[2]Barcelona Institute for Global Health, Hospital Clínic-Universitat de Barcelona, Barcelona, Spain
[3]Spanish Consortium for Research in Epidemiology and Public Health (CIBERESP), Madrid, Spain
[4]Department of Physiologic Sciences, Faculty of Medicine, Universidade Eduardo Mondlane, Maputo, Mozambique
[5]Malaria Consortium, London, UK
[6]Department of Medicine, University of California San Francisco, San Francisco, California, USA
[7]Bill & Melinda Gates Foundation, Seattle, Washington, USA
[8]Instituto Nacional de Saúde, Maputo, Mozambique
[9]Ministry of Health, Maputo, Mozambique

**Twitter** @genmoz_project

**Contributors** Conceived and designed the protocol: AM, CB, AR-F and BG. Gave inputs to protocol methodology: BC, CG, AC, ER-V, CdS, FS, SE, AW-S, PAr, SB, MR, NC, PAi and JP. Wrote the first draft of the manuscript: AM. Wrote, reviewed and approved the manuscript: all authors. Responsible for the overall content: AM.

**Funding** This work is supported by Bill and Melinda Gates Foundation (grant number INV-019032), including models and data analysis performed by the Institute for Disease Modeling at the Bill & Melinda Gates Foundation.

**Map disclaimer** The inclusion of any map (including the depiction of any boundaries therein), or of any geographic or locational reference, does not imply the expression of any opinion whatsoever on the part of BMJ concerning the legal

status of any country, territory, jurisdiction or area or of its authorities. Any such expression remains solely that of the relevant source and is not endorsed by BMJ. Maps are provided without any warranty of any kind, either express or implied.

**Competing interests** None declared.

**Patient and public involvement** Patients and/or the public were not involved in the design, or conduct, or reporting or dissemination plans of this research.

**Patient consent for publication** Not required.

**Provenance and peer review** Not commissioned; externally peer reviewed.

**ORCID iD**
Alfredo Mayor http://orcid.org/0000-0003-3890-2897

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
