## [Reviewer comments · BMJ Open]

ARTICLE DETAILS

TITLE (PROVISIONAL)	Protocol for a prospective surveillance study to detect antimalarial drug resistance, gene deletions of diagnostic relevance and genetic diversity of Plasmodium falciparum in Mozambique
AUTHORS	Mayor, Alfredo; da Silva, Clemente; Rovira-Vallbona, Eduard; Roca-Feltrer, Arantxa; Bonnington, Craig; Wharton-Smith, Alexandra; Greenhouse, Bryan; Bever, Caitlin; Chidimatembue, Arlindo; Guinovart, Caterina; Proctor, Josh; Rodrigues, Maria; Canana, Neide; Arnaldo, Paulo; Boene, Simone; Aide, Pedro; Enosse, Sonia; Saute, Francisco; Candrinho, Baltazar

VERSION 1 – REVIEW

REVIEWER	Fontecha, Gustavo Universidad Nacional Autonoma de Honduras Facultad de Ciencias
REVIEW RETURNED	18-Apr-2022

GENERAL COMMENTS	Some minor typing errors: Page 3 line 26 Page 7 line 20 Page 8 line 60 Page 9 lines 25 and 30 (italic) Page 10 line 7 (italic) Page 11 line 33 References 33 and 43 (WHO)
--

REVIEWER	Ferreira, Marcelo University of Sao Paolo
REVIEW RETURNED	07-May-2022

GENERAL COMMENTS	Mayor and colleagues provide a protocol of an ongoing, countrywide genomic surveillance study of Plasmodium falciparum malaria in Mozambique. The manuscript is clearly written and the study addresses a topic of major public health importance in a high-burden setting. Overall, study populations are well described and the different strategies to be used for recruiting participants in medium-high and low-endemicity settings are very appropriate. There are, however, two aspects of the study that are rather poorly described. The first key aspect to be described in greater is genome-wide genotyping methods and subsequent population genetic analysis. The authors aim to use an AmpliSeq approach to characterize polymorphisms in drug resistance markers, hrp2/hrp3, and to define "microhaplotypes", but do not provide an estimate of the number of SNPs to be characterized and do not describe their distribution across the parasite's chromosomes, the proportion of putatively neutral vs. nonneutral polymorphisms included in the
--

	assays. Moreover, the study protocol does not describe criteria for validating sequencing data (minimum sequencing depth, maximum error rate etc.). Second, the authors plan to analyze the "genetic connectivity" among isolates, but provide no clue regarding the methods to be used with this purpose. Do they plan to include all SNPs or only neutral SNPs in their analysis? How will "connectivity" be defined and detected? How do the authors plan to cope with multiple-clone infections in their analysis of genetic connectivity? How do they plan to integrate genetic data with "travel history" data? Which type of travel history data will be obtained from case notification records? In conclusion, this is a well-designed longitudinal study of genomic surveillance that will surely generate high-quality information for implementing improved malaria control measures in Mozambique, but some crucial aspects of a major component of the study -- namely, the population genomic analysis -- merits a more detailed description.
--	---

VERSION 1 – AUTHOR RESPONSE

Reviewer: 1

Some minor typing errors:

Page 3 line 26: Corrected

Page 7 line 20: Corrected

Page 8 line 60: Corrected

Page 9 lines 25 and 30 (italic): Corrected

Page 10 line 7 (italic): Corrected

Page 11 line 33: Corrected

References 33 and 43 (WHO): Corrected

Reviewer: 2

- There are, however, two aspects of the study that are rather poorly described. The first key aspect to be described in greater is genome-wide genotyping methods and subsequent population genetic analysis. The authors aim to use an AmpliSeq approach to characterize polymorphisms in drug resistance markers, *hrp2/hrp3*, and to define "microhaplotypes", but do not provide an estimate of the number of SNPs to be characterized and do not describe their distribution across the parasite's chromosomes, the proportion of putatively neutral vs. nonneutral polymorphisms included in the assays. Moreover, the study protocol does not describe criteria for validating sequencing data (minimum sequencing depth, maximum error rate etc.).

Response: We have not provided details about the genome-wide genotyping methods as the analytical approaches are not yet developed. An important part of the project is the development of these wet-lab approaches. For this reason, we cannot provide details in the protocol. While we have some preliminary data that supports the feasibility of our approaches, we prefer not to provide these details here, as this will be published in a specific research manuscript. We have clarified this in the manuscript as follows: "Genotyping typing methods, including number of SNPs and microhaplotypes to be characterized, distribution across the parasite's chromosomes, the proportion of putatively neutral vs. non-neutral polymorphisms, pooling strategy and criteria for validating sequencing data (i.e., minimum sequencing depth and maximum error rate) will be developed as part of this project."

- Second, the authors plan to analyze the "genetic connectivity" among isolates, but provide no clue regarding the methods to be used with this purpose. Do they plan to include all SNPs or only neutral

SNPs in their analysis? How will "connectivity" be defined and detected? How do the authors plan to cope with multiple-clone infections in their analysis of genetic connectivity? How do they plan to integrate genetic data with "travel history" data? Which type of travel history data will be obtained from case notification records?

Response: Similarly to the wet-lab approaches, the population genetic analysis and interpretative components are still under development as part of the project. We have also preliminary data and a path to build these approaches, but we prefer not to provide details in this manuscript that aims to reflect the protocol as approved by the ethic committees. We will provide methodological details in future papers. We have clarified this in the manuscript as follows: "Methods to be used for population genetic analysis, including the genetic connectivity among isolates, use of all versus only neutral SNPs, treatment of multiple-clone infections and integration of genetic data with "travel history" data) will be developed during the project".

With regards to travel history data collected during reactive surveillance strategies in low transmission settings in Maputo Province (Magde and Matutine Districts), travels during the previous 30 days to the case notification are registered, including destinations and dates.

VERSION 2 – REVIEW

REVIEWER	Ferreira, Marcelo University of Sao Paolo
REVIEW RETURNED	27-May-2022
GENERAL COMMENTS	The authors have addressed all comments and suggestions.